# A Novel Missense Variant Associated with A Splicing Defect in A Myopathic Form of PGK1 Deficiency in The Spanish Population

**DOI:** 10.3390/genes10100785

**Published:** 2019-10-10

**Authors:** Virginia Garcia-Solaesa, Pablo Serrano-Lorenzo, Maria Antonia Ramos-Arroyo, Alberto Blázquez, Inmaculada Pagola-Lorz, Mercè Artigas-López, Joaquín Arenas, Miguel A. Martín, Ivonne Jericó-Pascual

**Affiliations:** 1Department of Medical Genetics, Complejo Hospitalario de Navarra, IdiSNA, Navarra Institute for Health Research, 31008 Pamplona, Spain, ma.ramos.arroyo@navarra.es (M.A.R.-A.); merce.artigas.lopez@navarra.es (M.A.-L.); 2Laboratorio de Enfermedades Mitocondriales y Neurometabólicas. Instituto de Investigación Hospital 12 de Octubre, 28041 Madrid, Spain, pserranolorenzo.imas12@h12o.es (P.S.-L.); abencinar@hotmail.com (A.B.); joaquin.arenas@salud.madrid.org (J.A.); mamcasanueva.imas12@h12o.es (M.A.M.); 3Centro de Investigación Biomédica en Red de Enfermedades Raras (CIBERER), 28041 Madrid, Spain; 4Department of Neurology, Complejo Hospitalario de Navarra, IdiSNA (Navarra Institute for Health Research), 31008 Pamplona, Spain, inmaculada.pagola.lorz@navarra.es (I.P.-L.); ivonne.jerico.pascual@navarra.es (I.J.-P.)

**Keywords:** phosphoglycerate kinase 1 gene (*PGK1*), PGK1 deficiency, myopathic form, missense variant, abnormalities in mRNA splicing

## Abstract

Phosphoglycerate kinase (PGK)1 deficiency is an X-linked inherited disease associated with different clinical presentations, sometimes as myopathic affectation without hemolytic anemia. We present a 40-year-old male with a mild psychomotor delay and mild mental retardation, who developed progressive exercise intolerance, cramps and sporadic episodes of rhabdomyolysis but no hematological features. A genetic study was carried out by a next-generation sequencing (NGS) panel of 32 genes associated with inherited metabolic myopathies. We identified a missense variant in the *PGK1* gene c.1114G > A (p.Gly372Ser) located in the last nucleotide of exon 9. cDNA studies demonstrated abnormalities in mRNA splicing because this change abolishes the exon 9 donor site. This novel variant is the first variant associated with a myopathic form of PGK1 deficiency in the Spanish population.

## 1. Introduction

### 1.1. PGK Enzyme

The phosphoglycerate kinase (PGK) is a key enzyme for adenosine triphosphate (ATP) generation in the glycolysis. It catalyzes the reversible conversion of 1,3-bisphosphoglycerate (1,3-BPG) to 3-phosphoglycerate (3-PG), by the transfer of the high-energy phosphate from position 1 of 1,3-BPG to adenosine diphosphate (ADP) [1]. The 1,3-BPG may be metabolized to 2,3-bisphosphoglycerate (2,3-BPG) by bisphosphoglycerate mutase (BPGM), an enzyme only present in erythrocytes, and then the phosphate at position 2 is removed by bisphosphoglycerate phosphatase (BPGP) to 3-PG. The end product of both reactions is the same, i.e., 3-PG, but PGK also generates a molecule of ATP, while the direct glycolytic pathway through the 2,3-BPG occurs without any net gain of ATP, in what is known as “energy clutch” and was described by Rapoport and Luebering in 1950 [2]. The regulation of the metabolism at this point determines not only ATP production but also the concentration of 2,3-BPG, an important regulator of the oxygen affinity of hemoglobin and its release in tissues.

PGK was also shown to participate in DNA replication and repair in mammal cell nuclei [3]. Extracellular PGK has been reported to exhibit thiol reductase activity on plasmin, which inhibits tumor angiogenesis [4,5]. In addition to the phosphotransferase activity in the glycolysis pathway, PGK can phosphorylate L-nucleoside analogues, used in antiviral and anticancer therapies, to the respective active triphosphates [6,7]. 

There are two isoenzymes of PGK with a similar structure and function but different tissue expression and encoded by different genes with different genetic origins: PGK1, expressed in all somatic cells except spermatogenic cells; and PGK2 expressed only in the late stages of spermatogenesis [1,8,9].

### 1.2. PGK1 Deficiency

PGK1 deficiency is an X-linked inherited disease (OMIM# 300653); the human PGK1 gene contains 11 exons and spans approximately 23 kilobases in Xq21.1. Despite PGK1 being a ubiquitous enzyme, the clinical presentation of PGK1 deficiency depends on three tissues: Erythrocytes (anemia), skeletal muscle (myopathy), and the central nervous system (CNS) [10,11]. It is generally associated with moderate to severe non-spherocytic hemolytic anemia, often accompanied with CNS involvement [8]. In some cases, PGK1-deficient patients exhibit only muscular symptoms without anemia. Mental retardation, behavioral abnormalities, seizures or strokes represent the main neurological alterations, whereas cramps and myoglobinuria characterize the myopathic forms that may lead to acute renal failure. In fact, PGK1 deficiency is one of the less frequent types of muscle glycogenoses or Glycogen Storage Diseases (GSD), specifically, type IX (GSD-IX), and, as such, these patients present similar clinical features to those reported in other muscle glycogen storage disorders [10].

Since the first family described in 1968 [12], nearly another 33 families affected by PGK1 deficiency have been reported. In total, 29 of them have been characterized at the molecular level, with a total of 25 different mutations identified, and two splicing defects have been described without the identification of the genetic change [13,14]. Furthermore, another two variants known as PGK München [15,16] and PGKII or PGK Samoa have been identified as polymorphisms [17,18] (Table 1).

Most of the mutations reported are missense (18). There are five splicing mutations, two small deletions (one of them is a frameshift deletion) and one small indel (Table 1). Although nearly all mutations have been characterized at the level of enzymatic activity and protein stability, correlation between genotype and phenotype based on the location of mutations within the different protein domains or in critical regions for protein activity has not been established [19]. No pattern has emerged that would explain the different clinical manifestations of the various mutations [1].

## 2. Materials and Methods

### 2.1. Case Report

A 40-year-old male was referred to the Department of Neurology of Navarra’s Hospital after suffering two episodes of rhabdomyolysis in the last 2 years without an apparent trigger, complicated with acute renal failure, requiring hemodialysis. He was the eldest of two siblings born to non-consanguineous parents of Spanish origin and was studied in childhood for mild psychomotor delay with frequent falls and a first episode of rhabdomyolysis triggered by exercise (learning to swim) at the age of seven without reaching a specific diagnosis (we have not been able to access previous studies that included a muscle biopsy). From adolescence, he developed a progressive clinical picture of fatigue, exercise intolerance, cramps and recurrent episodes of myoglobinuria. No “second wind” phenomenon was reported and there was no previous history of seizures or other central nervous system (CNS) dysfunctions. Laboratory studies demonstrated increased resting serum creatine kinase levels (x7–10) and normal red cell count in consecutive analyses.

At the time of our neurological evaluation, the physical examination revealed mild cognitive disability, global muscle amyotrophy, facial weakness with bilateral ptosis, proximal and symmetrical weakness in the limbs and neck flexors, without scapular winging or contractures. No evidence of Parkinsonism or tremor. Cerebral and muscle magnetic resonance imaging, abdominal ultrasound and echocardiogram were normal. The analysis showed a normal acylcarnitine profile and reduced lactate production in exercise test. On suspicion of metabolic myopathy, in particular a muscle glycogenoses, a biceps brachialis muscle biopsy was performed without showing significant findings. No vacuoles or glycogen accumulation were observed, and phosphorylase activity was normal. A metabolic myopathies panel was performed.

### 2.2. Molecular Genetic Studies

#### 2.2.1. Metabolic Myopathies Panel

This study was performed in accordance with the declaration of Helsinki and was approved by the Ethics Committee of Complejo Hospitalario of Navarra. Patient inclusion and case publication were approved by the Ethics Committee of Complejo Hospitalario of Navarra. Proyect 2017/68. Blood was collected from the patient and relatives after obtaining informed consent. DNA was extracted using a MagNa Pure Compact Nucleic Acid Isolation Kit I and MagNA Pure Compact Instrument (Roche Molecular Diagnostics, Pleasanton, CA, USA).

A customized next-generation sequencing (NGS) panel of 32 genes associated with inherited metabolic myopathies was designed using AmpliseqTM and sequencing with the PGM-Ion Torrent platform (Life Techonologies) (Appendix A). The alignment of the sequences (ref. CRCh37/hg19) and detection of variants was performed in Torrent Suite (TMAP-variantCaller plugin). The annotation and prioritization of variants has been carried out through the integration of own scripts with Annovar [47]. The theoretical coverage of this panel reaches 99.21%. This analysis did not cover all of target regions due to characteristics inherent to this methodology.

#### 2.2.2. Clinical and Methodological Validation

Variant prioritization was performed assuming an autosomal recessive or recessive X-linked inheritance following the next steps: (i) Minor Allele Frequency (MAF), <0.05 in population database, including 1000 Genomes Project (http://browser.1000genomes.org), Exome Variant Server (http://evs.gs.washington.edu/EVS) or Exome Aggregation Consortium (ExAC) (http://exac.broadinstitute.org/), Genome Aggregation Database (gnomAD) (https://gnomad.broadinstitute.org), Single Nucleotide Polymorphism database (dbSNP) (http://www.ncbi.nlm.nih.gov/snp), and Collaborative Spanish Variant Server (http://csvs.babelomics.org); (ii) intronic variants localized far from 15 nucleotides of the exon/intron junction were discarded; iii) status and ranking of the variants in the ClinVar database (http://www.ncbi.nlm.nih.gov/clinvar); (iv) variant pathogenicity predictors including Sorting Intolerant from Tolerant (SIFT) (http://sift.jcvi.org), PolyPhen-2 (http://genetics.bwh.harvard.edu/pph2, Likelihood Ratio Test (LRT), MutationTaster (http://www.mutationtaster.org), Mendelian Clinically Applicable Pathogenicity Score (M-CAP) (http://bejerano.stanford.edu/mcap/), Protein Variation Effect Analyzer (PROVEAN) (http://provean.jcvi.org/index.php) and Combined Annotation Dependent Depletion (CADD) Phred (http://cadd.gs.washington.edu), Human Splicing Finder; (v) assessment of phylogenetic conservation using Genomic Evolutionary Rate Profiling (GERP), and the Phylogenetic Analysis with Space/Time models (PHAST) programs: phastCons and phyloP.

Methodological validation and family segregation studies of the new variant were performed by direct sequencing (ABI 3500 Genetic Analyzer, Applied Biosystems, Warrington, UK) using a Big Dye Terminator Cycle Sequencing Kit (Applied Biosystems, Warrington, UK). The chromatograms were analyzed with Chromas 2.3 (Technelysium Pty Ltd.).

Available databases were used to determine the complete sequence of the gene and design the most appropriate primers for the PCR: University of California Santa Cruz USCS Genome Browser (https://genome.ucsc.edu/) and Primer3plus (http://www.bioinformatics.nl/cgi-bin/primer3plus/primer3plus.cgi/). The sequences of the upstream and downstream primers used to amplify the genomic region surrounding the variant of interest located in exon 9 of the *PGK1* gene were 5′-GGTCCTGAAAGCAGCAAGAA-3′ and 5′-CTCCCCAACCCAAAAGGTAG-3′, respectively. 

#### 2.2.3. cDNA Analysis of *PGK1*

Peripheral blood mononuclear cells (PBMCs) were isolated from whole blood of the index case and a healthy control by Ficoll gradient centrifugation according to the manufacturer’s instructions (Ficoll-Paque PLUS, GE Healthcare).

Total RNA was isolated from PBMCs using TRIzol Reagent (Invitrogen, Thermo Fisher Scientific). The cDNAs were synthesized from 1 μg of total RNA using random hexamers with the SuperScript IV-First Strand Synthesis System kit (Invitrogen, Thermo Fisher Scientific, Waltham, MA, USA) in a total volume of 20 μL.

Subsequently, cDNA of *PGK1* was amplified and Sanger sequenced using specific primers surrounding the exon 9 of *PGK1*. The sequence of the upstream primer, located in the exon 7–8 junction of *PGK1*, was 5′-GTGCTCAACAACATGGAGAT-3′, and the sequence of the downstream primer located inside the exon 11 of *PGK1* was 5′-TAAATATTGCTGAGAGCATCCA-3′. Amplicons were visualized on a 1% agarose gel using gel red and were purified from the gel with IllustraTM GFXTM PCR DNA and Gel Band Purification Kit (GE Healthcare) to be analyzed separately by direct sequencing (ABI 3500 Genetic Analyzer, Applied Biosystems, Warrington, UK) using a Big Dye Terminator Cycle Sequencing Kit (Applied Biosystems, Warrington, UK). The chromatograms were analyzed with Chromas 2.3 (Technelysium Pty Ltd.).

### 2.3. Enzyme Activity of PGK1

The blood enzymatic activity of PGK was analyzed by molecular absorption spectrometry. The enzymatic analysis of RBCs was performed as described previously by the method of the International Committee for Standardization in Hematology (ICSH) [48].

## 3. Results

To identify the genetic cause of the patient’s myopathy, we performed a customized NGS panel including 32 genes associated with metabolic myopathies. We found 174 variants—Of which, only one variant was prioritized after applying the established criteria, a hemizygous c.1114G > A substitution that predicts a (p.Gly372Ser) missense mutation in the *PGK1* gene (NM_000291). This variant was absent in the gnomAD and the 1000 genomes databases and was categorized as deleterious by in silico predictors such as SIFT, PolyPhen2, LRT, PROVEAN, MutationTaster, CADDPhred, GERP and PhyloP. The variant was registered in the ClinVar database as an uncertain significance (VUS) [49]. Following the American College of Medical Genetics and Genomics (ACMG) guidelines for the interpretation of sequence variants [50], the variant was classified as likely pathogenic.

The Gly-372 residue is located adjacent to the catalytic site of the enzyme [19] and is highly conserved in this protein family among different species (Figure 1A). Moreover, the variant falls in the last nucleotide of 3′ at the end of exon 9, so that in silico analysis using Human Splice Finder and Mutation Taster also suggested that the variant would alter the splicing process, since this change abolishes the exon 9 donor site. 

The presence of the hemizygous variant c.1114G > A (p.G372S) in the *PGK1* gene in the patient was confirmed by Sanger sequencing. The asymptomatic parents were analyzed for the variant, which was found to be heterozygous in the patient’s mother and absent in the father (Figure 1B and Figure 2).

The analysis of the patient’s blood cDNA revealed the presence of two abnormal transcripts: (i) One apparently more abundant species showing a skipping of the exon 9, and (ii) a barely detectable band corresponding to the transcript with a retention of intron 9 (Figure 3). cDNA analysis also reveals that normal splicing is present in the patient, although the missense variant is observed in the sequence of this transcript.

The enzyme activity of PGK was decreased in the patient’s blood, 39 UI/g Hb (normal range 197–343).

## 4. Discussion

We identified a novel hemizygous missense variant associated with PGK1 deficiency, c.1114G > A (p.G372S) in a male adult patient presenting with exercise intolerance and several episodes of rhabdomyolysis, ptosis, muscle weakness and mild cognitive disability.

The variant is probably the cause of the clinical phenotype, since (i) it was the only prioritized variant after analyzing a customized NGS panel including 32 genes associated with metabolic myopathies; (ii) it was absent in several population databases; (iii) and although it was found as a VUS in the ClinVar database, the ACMG guidelines classified the variant as ‘likely pathogenic’; (iv) the amino acid residue is located adjacent to the catalytic site of the enzyme and is highly evolutionary conserved; (v) the 11/11 predictors of pathogenicity indicated the variant is deleterious; (vi) the variant was located at the last nucleotide of the 3′ end of exon 9, and predictors of aberrant splicing suggested that the variant affects mRNA maturation, and this fact was demonstrated by the patient’s cDNA studies (Figure 3); (vii) the PGK enzyme activity in the patient’s blood cells was strongly decreased.

The variant c.1114G > A (p.G372S) is the first mutation associated with a myopathic form of PGK1 deficiency in the Spanish population; two other Spanish patients who were previously reported with PGK1 deficiency with different mutations in the PGK1 gene presented hemolytic anemia and CNS involvement, with no signs of myopathy (Table 1).

The c.1114G > A variant in our patient, despite being located in the last nucleotide of the 3′ end exonic region of exon 9 of the PGK1 gene, was demonstrated to affect mRNA maturation by cDNA studies on the patient’s blood cells. We showed that the mutation c.1114A > G leads to the expression of an apparent main aberrant cDNA-PGK1 species in which exon 9 skipping occurred. In addition, besides the canonical transcript carrying the predicted missense mutation, another longer abnormal transcript was detected which includes a retention of the entire intron 9 of the gene (Figure 3). In summary, c.1114G > A that was initially predicted to lead to a missense change, actually, also gave rise to a splicing defect, resulting in two structural aberrant transcripts of the PGK1 gene that generate a premature stop codon (PTC) (p.G313Vfs * 20 due to exon 9 skipping and p.G372Gfs * 11 due to intron 9 retention).

There are five other demonstrated splicing variants of PGK1 that have been reported so far, namely, IVS4 + 1G > T (North Carolina), p.Gly213=, p.Glu252Ala (Antwerp), IVS7 + 3A > G and IVS7 + 5G > A (Fukuroi) [25,36,37,39,40,41]. In addition, two splicing defects have been described lacking the identification of the genetic mutation [13,14]. All of these cases were reported in patients with the myopathic form of PGK1 deficiency and with no hemolytic anemia, in a similar manner to our patient. However, there are five other patients with the myopathic phenotype—Four of them were carriers of missense mutations, namely, p.Cys108Tyr, p.Ile253Thr (Hamamatsu), p.Asp315Asn (Creteil) and p.Thr378Pro (Afula) [19,24,28,42]. The fifth was a male hemizygous for a frameshift variant with a four base pair deletion in exon 6 that predicts the formation of a truncated protein (p.Gly213Glufs * 21, Fukui) [35]. Interestingly, three of these missense mutations (Hamamatsu, Creteil and Afula) are located at the beginning or at the end of the corresponding exons similarly to the mutation described here (Figure 4) but a splicing defect was not experimentally discarded, as it was ruled out for other previously reported missense variants localized in these critical regions for mRNA splicing [11].

Regarding genotype–phenotype correlation in PGK1 deficiency, all the mutations that affect splicing have been associated with myopathic forms and absence of hemolytic anemia. Although we cannot state that all myopathic forms without anemia are associated with splicing variants, it seems conceivable to consider this hypothesis, since some reported missense mutations located in critical exonic regions for splicing were not discarded as being involved in the abnormal splicing of PGK1. This is no new concept, because a large and growing number of variants located in protein coding exons have primary disease-causing effects by disrupting splicing—For example, Spinal Muscular Atrophy (SMA) is caused by the loss of *SMN1* genes and the C to T substitution in exon 7 of the *SMN2* gene that promotes exon skipping in *SMN2* and could not compensate the loss of SMN1 [51].

A patient recently reported to harbor a missense variant of PGK1 (c.649G > A, p.Val217Ile) and intellectual disability, mild cerebral and cerebellar atrophy and peculiar episodes of muscle weakness of unknown etiology, but without hemolytic anemia, had a residual enzymatic activity of 78–91% in RBCs [38]. This finding would suggest this missense variant, apparently not involved in splicing, is a neutral polymorphism.

The correlation between variants that apparently are the most damaging at the protein level such as splicing or frameshift and milder forms of PGK1 deficiency are more complicated to explain; in fact, the levels of residual enzyme activity in these patients are very low (Table 1), and it was shown that these individuals have a much lower amount of enzyme. Recent studies of RNA massive sequencing indicate that the number of events of alternative splicing and produced isoforms are much more abundant than had been estimated previously [52], and that alternative transcripts of PGK1 could even exist that have not been detected to date. This mechanism could explain the phenotype of the frameshift deletion (Fukui variant), where a third part of the protein is missing but surprisingly the mutation is compatible with life and enzymatic activity is detected in erythrocytes and muscle.

In addition, two cases, i.e., PGK Antwerp and our patient, both harboring a predicted missense variant that leads to a combined alteration of the splicing process and the expression of a mutated canonical transcript, showed only less severe myopathic forms. Moreover, in our patient, we showed that the canonical transcript harbors the change c.1114G > A, which is predicted to substitute a Gly residue by Ser in position 372 of the enzyme that is localized adjacent to the catalytic site. Therefore, it would be expected to express a more severe phenotype. Although, it has still to be characterized at the enzymatic level, it is known that G372, G373 and G374 are important residues in the enzyme, coordinating the nucleotide substrate and stabilizing contacts with G395, thus maintaining the closed catalytically active conformation [11].

Previously, it has been attempted to associate intermediate metabolites accumulated in PGK-deficient red cells with hemolysis. These intermediaries, such as 2,3-BPG, inhibit different enzymes of glycolysis and others pathways, such as 6-phosphogluconate dehydrogenase [37], promoting hemolysis. A significant increase in 2,3-BPG in red cells was described in a few patients with hemolytic anemia related to PGK deficiency (PGK Barcelona, PGK Matsue, PGK Amiens/New York and PGK Uppsala), because it has not been quantified in most of the described cases. However, in PGK Creteil, which is not associated with hemolytic anemia, a moderate increase in 2,3-BPG levels in RBCs was described [9]. Perhaps the correlation of the phenotype should not be established with the enzyme activity itself, but with the stability of the protein and its tendency to form aggregates as already raised in 2014 by Pey et al. [53]. From the variants studied by this group, those causing severe affectation to protein stability and mild catalytic efficiency (Barcelona, Matsue, Michigan, Murcia and Kyoto) were associated with hemolytic anemia and neurological disorders but not myopathy. We propose that the aggregates themselves could accumulate and cause damage in different tissues, the in case of RBCs in the form of hemolytic anemia. The splicing or deletion variants would protect from this toxic deposit, which would explain why none of the described splicing variants are associated with hemolytic anemia. In these patients, the clinical picture resembles that of muscular glycogenosis—in which, the symptoms are exacerbated with exercise due to the inability of the mutant enzyme to cope with energy demand. The limitation in establishing this new idea about the phenotype–genotype association may be due to the fact that we could not check how the p.Gly372Ser missense variant affects stability and/or enzymatic activity in vitro and also the variability of the results described so far in terms of the methods used for the identification of variants and functional studies of enzymatic activities. Some of these studies are from 50 years ago; the first case of PGK deficiency was described in 1968 [12]. 

From the point of view of CNS involvement, the phenotype is heterogeneous. Most of the myopathic forms without hemolytic anemia are associated with mild mental retardation but not with other neurological manifestations. However, the forms characterized by hemolytic anemia are described in patients with a wide range of neurological disturbances ranging from no neurological problem to mental retardation, Parkinson’s [1] or retinal dystrophy [31]. Considering the muscular symptoms, phenotypic heterogeneity is also found. Myopathy may be underestimated in the cases described at an earlier age due to its later onset. 

However, although the mechanism underlying this clinical heterogeneity of PGK1 deficiency remains unknown, we can conclude with dichotomization in myopathic forms without anemia and hemolytic forms. Finally, we highlight that patients with myopathic forms of PGK1 deficiency might be underdiagnosed due to the absence of hemolytic anemia.

## Figures and Tables

**Figure 1 genes-10-00785-f001:**
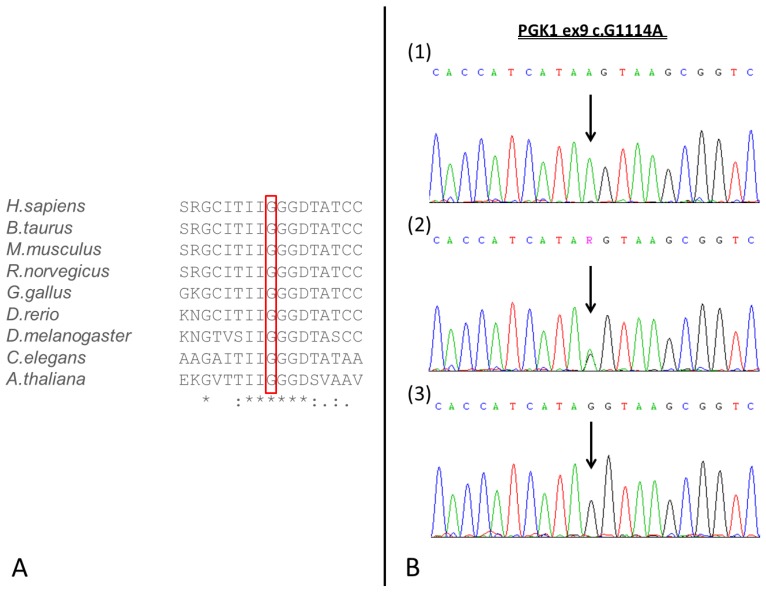
(**A**) Evolutionary conservation of the G372 residue among different species. Asterisk, fully conserved residue; colon, conservative replacement; period, semi-conservative replacement. (**B**) DNA sequence chromatograms. The picture shows the sequencing results of the patient and his parents from the genomic DNA variant c.1114G > A (p.G372S) in the PGK1 gene. Hemizygous index case c.1114A (1b), heterozygous mother c.1114GA (2b) and wild type father (3b).

**Figure 2 genes-10-00785-f002:**
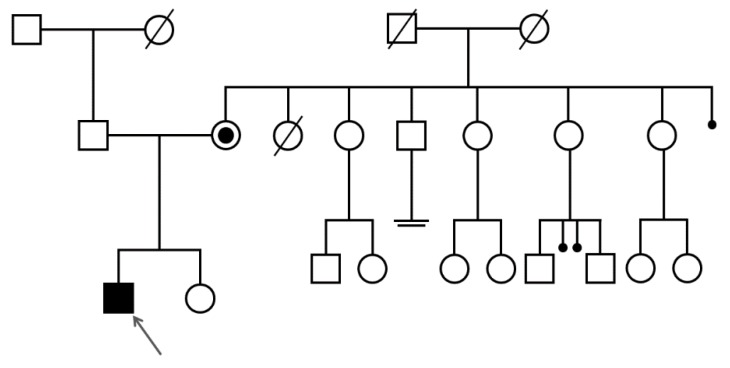
Family pedigree illustrating the patient’s family, showing that the patient mother is a carrier of the variant. The four sisters of the patient’s mother and the patient’s sister have not been analyzed for their carrier status.

**Figure 3 genes-10-00785-f003:**
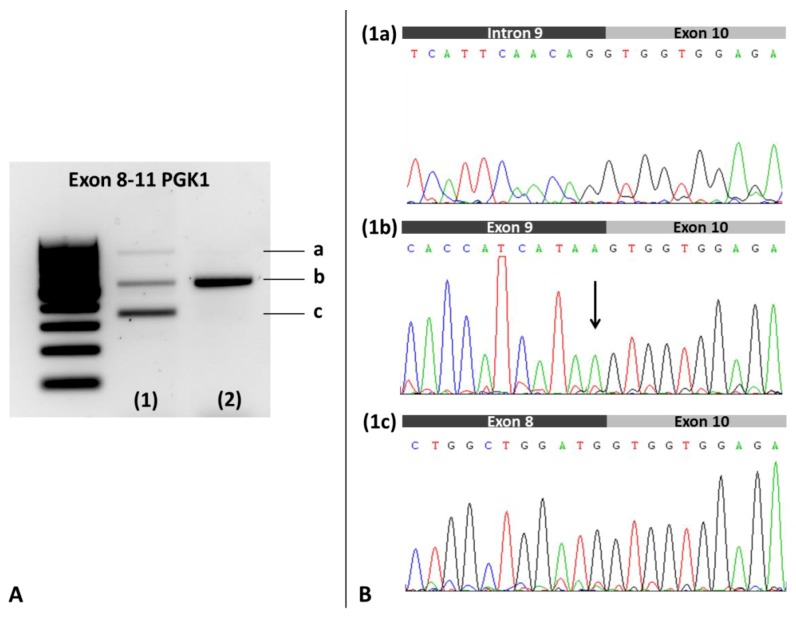
Analysis of the PGK1 patient’s blood cDNA. (**A**). Electrophoresis on 1% agarose gel. Lane 1: Index case showing a transcript of normal size (b band), one higher molecular weight cDNA species (a band), and an apparently more abundant transcript of lower molecular weight (c band); Lane 2: Healthy control showing a transcript of normal size (b band). (**B**). Sanger Sequencing of the gel purified amplicons. (**1a**): Sequence of patient’s a band showing the retention of intron 9; (**1b**): Sequence of the patient’s b band displaying the junction of exons 9 and 10, and the variant c.1114G > A (indicated by an arrow). Healthy control revealed the same pattern for b band (not shown). The hemizygous c.1114G > A variant was detected in the patient’s sequence of a transcript of normal size (1b); (**1c**): Sequence of patient’s c band showing the skipping of exon 9 in the cDNA.

**Figure 4 genes-10-00785-f004:**
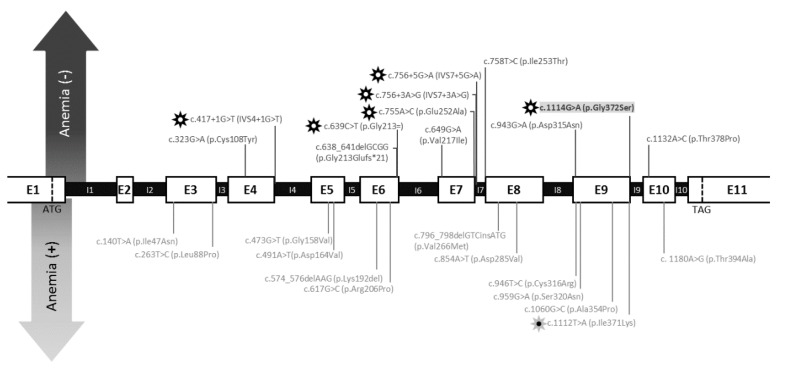
Schematic representation of the described mutations in the PGK1 gene, represented by separate mutations associated with hemolytic anemia and mutations present in non-hemolytic forms. (Black star: mutations affecting splicing. Grey star: mutation in which it has been ruled out that the splicing is altered).

**Table 1 genes-10-00785-t001:** Phosphoglycerate kinase (PGK)1 mutations and clinical symptoms.

Variant	Age of Diagnosis	Age of Last Review	PGK Residual Activity (%)		Nucleotide Change	Amino Acid Change	Symptoms			Comments on The Studies: Phenotype or Methodology
			RBC	Muscle			H	M	C N S	
Barcelona [9,20]	3	7	10.4/20	NA	c.140T > A	p.Ile47Asn	+	-	+	Neonatal anemia (Hb: 7.3g/dL) and progressive neurological impairment leading to mental retardation (7 years). The decrease in PGK activity is more closely related to a loss of enzyme stability than to a decrease in catalytic function.
Matsue [21,22,23]	9	9 †	5/10	NA	c.263T > C	p.Leu88Pro	+	-	+	Higher Michaelis-Menten constant (Km) for all substrates, particularly for ATP and 1,3-BPG
[24]	32	NA	1.9	2.6	c.323G > A	p.Cys108Tyr	-	+	+	Chronic axonal sensorimotor polyneuropathy and mental retardation, microcephaly and ophthalmoplegia.
North Carolina [25]	12	NA	2.7	2	c.417 + 1G > T	IVS4 + 1G > T	-	+	+	Mild intellectual delay with attention-deficit disorder.
Shizuoka [26]	27	NA	0.7	NA	c.473G > T	p.Gly158Val	-/+	+	-	No neonatal hemolytic anemia (Hb: 12.8 g/dL from diagnosis).
Amiens/New York [27,28,29,30,31]	0/3	23	<5		c.491A > T	p.Asp164Val	+	-	+	Seizures associated with hemolytic anemia. Retinal dystrophy. The most deleterious variant was at the protein level.
Alabama [32]	37	NA	4	NA	c.574_576delAAG	p.Lys192del	-/+	-	-	School teacher who has been in excellent general health. At age 20, he had a self-limited febrile illness associated with a high bilirubin level.
Uppsala [33,34]	26	31	10	NA	c.617G > C	p.Arg206Pro	+	-	+	Anemia and jaundice since 4 months after birth. Lower affinity for substrates, ATP and 1,3-BPG. Significant accumulation of 2,3-BPG and 2-phosphoglycerate (2-PG).
Fukui [35]	36	NA	5.6	2.9	c.638_641delGCGG	p.Gly213Glufs* 21	-	+	-	Complementary DNA sequence of a reverse transcriptase PCR product of exon 6 in the *PGK1* gene from leukocytes. No more information about methodology.
[36,37]	16/21	NA	4–5	2–3	c.639C > T	p.Gly213 =	-	+	-	Two brothers with normal intelligence.
[38]	16	NA	78–91	NA	c.649G > A	p.Val217Ile	-	+	+	Polymorphism: Male with intellectual disability, epileptic seizures, mild cerebral and cerebellar atrophy and peculiar episodes of muscle weakness of unknown etiology.
Antwerp [39]	25	NA	5.6	8	c.755A > C	p.Glu252Ala	-	+	-	Slightly decreased hemoglobin (13.2 g/100 mL) (normal: 13.3).
[40]	14/16	NA	<15	NA	c.756 + 3A > G	IVS7 + 3A > G	-	+	+	Mild intellectual deficiency. At 8 years old, after 20 min in a swimming pool, one of them had myoglobinuria.
Fukuroi [41]	33	NA	13.6	8.9	c.756 + 5G > A	IVS7 + 5G > A	-	+	+	A 33-year-old man first had severe muscle pain and myoglobinuria after a short run at age 20 years. No history of epileptic attacks despite small spike waves on electroencephalogram (EEG).
Hamamatsu [42]	11	NA	8.2	4.4	c.758T > C	p.Ile253Thr	-	+	+	PGK1 mRNA was reverse transcribed and amplified in three fragments and subcloned and sequenced.
Tokyo [18]	NA	NA	10	NA	c.796_798delGTCinsATG	p.Val266Met	+	-	+	Lower specific activity and increased thermal instability.
München [15,16]	polymorphism				c.802G > A	p.Asp268Val	-	-	-	Associated with enzyme deficiency and heat instability
Herlev [43]	69	72 †	49	NA	c.854A > T	p.Asp285Val	-/+	-	-	Pronounced reticulocytosis (10–45%). Approximately 90% of the mutated nucleotide T, approximately 10% of normal A nucleotide. Mosaicism? Somatic mutation?
Creteil [28]	31	NA	3	25	c.943G > A	p.Asp315Asn	-	+	-	Since his childhood, he presented several symptoms during physical exercises, notably rhabdomyolysis crises,
Michigan [44]	9	14	10	NA	c.946T > C	p.Cys316Arg	-/+	-	+	Compensated hemolytic anemia with occasional hemolysis crises (infections). The variant was more labile than the normal enzyme. De novo variant.
Murcia [20]	6	7†	49.2	NA	c.959G > A	p.Ser320Asn	+	-	+	Required transfusions from birth every 3–4 weeks. Cortical and subcortical atrophy.
PGKII/Samoa [17,18]	polymorphism				c.1055C > A	p.Thr352Asn	-	-	-	Electrophoretic variant not associated with enzyme deficiency.
Kyoto [45]	3	3.2	6.3	NA	c.1060G > C	p.Ala354Pro	+	+	+	Anemia and jaundice at birth. Respiratory infection-associated hemolytic crisis and rhabdomyolysis during early infancy.
[46]	4	25	14.6	NA	c.1112T > A	p.Ile371Lys	+	+	+	At the age of 25 years, he shows generalized myopathy, intelligence quotient (IQ)52 and cerebellar atrophy.
Present study	38	40	19.7	NA	c.1114G > A	p.Gly372Ser	-	+	+	Mild intellectual deficiency. Progressive exercise intolerance, cramps and sporadic episodes of rhabdomyolysis.
Afula [19]	18/25	NA	2	0.9/1.1	c.1132A > C	p.Thr378Pro	-	+	-	(18) For 7 years, he has experienced recurrent episodes of muscle cramps, myalgia and pigmenturia after intense exercise.(25) He had severe parkinsonism that was responsive to levodopa.

H: Hemolytic anemia. M: Myopathy. CNS: Central Nervous System affectation. †: Patient died (age); +: presence; -: not presence; -/+: moderate anemia (Shizuoka) or hemolysis well compensated (Alabama, Michigan). RBC: red blood cells. Hb: hemoglobin. NA: Not available.

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
