# Peer review of "A Novel Missense Variant Associated with A Splicing Defect in A Myopathic Form of PGK1 Deficiency in The Spanish Population"

_genes, 2019, doi:10.3390/genes10100785_

Round 1

Reviewer 1 Report

The authors present a 40 year-old male with inherited metabolic myopathie. Genetic study was carried out by a next-generation sequencing  panel of 32 genes associated with inherited metabolic myopathies. They identified a missense variant  in the PGK1 gene; c.1114G>A (p.Gly372Ser), located in the last nucleotide of exon 9.

The study is well designed and carried out. The applied next-generation sequencing is appropriate for obtaining trustworthy results.

Comments

The description of patient is insufficient and should be extended.  

Table 1 is unclear and should be corrected.

Please provide the list of 32 genes that were associated with inherited myopathies, that were used in the current study as potential targets (possibly as supplementary file).

In Discussion the authors should explain the limitations of the study and provide explanation for discrepancies.

Author Response

Reviewer 1

Comments and Suggestions for Authors 

The authors present a 40 year-old male with inherited metabolic myopathy. Genetic study was carried out by a next-generation sequencing panel of 32 genes associated with inherited metabolic myopathies. They identified a missense variant in the PGK1 gene; c.1114G>A (p.Gly372Ser), located in the last nucleotide of exon 9.

The study is well designed and carried out. The applied next-generation sequencing is appropriate for obtaining trustworthy results.

Comments

The description of patient is insufficient and should be extended

Thank you for your comments. We have now expanded the patient-related information (lines 84-104)

Table 1 is unclear and should be corrected.

Thank you for your comments. We have now explained better the table 1

Please provide the list of 32 genes that were associated with inherited myopathies, that were used in the current study as potential targets (possibly as supplementary file).

Thank you for your comments. We have now added the genes included in the NGS panel as supplementary file (Table S1).

In Discussion the authors should explain the limitations of the study and provide explanation for discrepancies.

Thank you for your comments. We have added the limitation of the study (lines 315-320).

Reviewer 2 Report

The manuscript by Garcia-Soleasa et al. describes a case of PGK1 splicing defect in a patient presenting a myopathic form of PGK1 deficiency. The authors present the variant, confirm the splicing alteration by RT-PCR and enzymatic reduction by enxymatic activity assay.

The description is interesting however the manuscript should be either converted in a review of the literature (as for table 1) adding this patient or in a case report, thereby reducing introduction and discussion sections.

Major points:

The manuscript needs a thorough revision of English as there are many grammar and syntax mistakes in the text. The title should be corrected “this novel variant…. In Spanish population” does not reinforce the study. With rare disease, it is common to have few patients, molecularly distinct in different areas.

Minor points:

Figure 1 should be removed. “serum CK (x7-10)” line 86. Please explain Line 96. The NGS panel is introduced. However, the 32 genes are never detailed. Moreover, results of NGS are not presented. Line 148. Please add detail to this method paragraph.

Author Response

Comments and Suggestions for Authors

The manuscript by Garcia-Solaesa et al. describes a case of PGK1 splicing defect in a patient presenting a myopathic form of PGK1 deficiency. The authors present the variant, confirm the splicing alteration by RT-PCR and enzymatic reduction by enzymatic activity assay.

The description is interesting however the manuscript should be either converted in a review of the literature (as for table 1) adding this patient or in a case report, thereby reducing introduction and discussion sections.

Thank you for your comments. We have presented the paper as an article because the review was made to raise a genotype-phenotype association of our case that had not been done so far in the literature, in which all forms of splicing are associated with myopathic forms of PGK1 deficiency.

Major points:

The manuscript needs a thorough revision of English as there are many grammar and syntax mistakes in the text. The title should be corrected “this novel variant…. In Spanish population” does not reinforce the study. With rare disease, it is common to have few patients, molecularly distinct in different areas.

Thank you for your comments. The article has undergone English language editing by MDPI. The text has been checked for correct use of grammar and common technical terms, and edited to a level suitable for reporting research in a this journal.

We decide to change the title: A novel missense variant associated with a splicing defect in a myopathic form of PGK1 deficiency in the Spanish population

Minor points:

Figure 1 should be removed.

Done

“serum CK” line 86.

Serum creatine kinase (x7-10) (corrected)

Please explain Line 96. The NGS panel is introduced. However, the 32 genes are never detailed. Moreover, results of NGS are not presented.

Thank you for your comments. We have now added the genes included in the NGS panel as supplementary file (Table S1). We indicated in the methods section the prioritization criteria used. Besides, in the results section we state that applying those criteria, we prioritized only 1 out of 174 variants. Complete list of variants available on request.

Line 148. Please add detail to this method paragraph.

Thank you for your comments. The methodology used for the quantification of the PGK activity is the same as that contained in the article mentioned: spectrophotometry measurement at 340nm of colorimetric changes resulting from the reduction reaction of a substrate mediated by the enzyme, at 37ºC. The reference ranges, as also mentioned in the article, they should be established at each laboratory by calculating the mean of the normal population ±2SD. (lines 168-170 and reference 48)

Round 2

Reviewer 2 Report

After revision, the manuscript has improved